# On the connection between real-world circumstances and online player behaviour: The case of EVE Online

**Andres M. Belaza**[1,2]*, **Jan Ryckebusch**[1], **Koen Schoors**[2,3], **Luis E. C. Rocha**[1,2], **Benjamin Vandermarliere**[1,2]

**1** Department of Physics and Astronomy, Ghent University, Ghent, Belgium, **2** Department of Economics, Ghent University, Ghent, Belgium, **3** National Research University, Higher School of Economics, Moscow, Russia

* andres.belaza@ugent.be

**Data Availability Statement:** All relevant data for the country profiles are within the paper and its Supporting Information files. The data sets for the in-game activities of the players of EVE online belong to Crowd Control Productions (URL:

## Abstract

Games involving virtual worlds are popular in several segments of the population and societies. The online environment facilitates that players from different countries interact in a common virtual world. Virtual worlds involving social and economic interactions are particularly useful to test social and economic theories. Using data from EVE Online, a massive online multi-player game simulating a fantasy galaxy, we analyse the relation between the real-world context in which players live and their in-game behaviour at the country level. We find that in-game aggressiveness to non-player characters is positively related to real-world levels of aggressiveness as measured by the Global Peace Index and the Global Terrorist Index at the country level. The opposite is true for in-game aggressiveness towards other players, which seems to work as a safety valve for real-world player aggressiveness. The ability to make in-game friends is also positively related to real-world levels of aggressiveness in much the same way. In-game trading behaviour is dependent on the macro-economic environment where players live. The unemployment rate and exchange rate make players trade more efficiently and cautiously in-game. Overall, we find evidence that the real-world environment affects in-game behaviour, suggesting that virtual worlds can be used to experiment and test social and economic theories, and to infer real-world behaviour at the country level.

## Introduction

Virtual worlds are computer-simulated games where real-world people or players can create avatars to live, interact and communicate among themselves or with the virtual environment within a particular context or theme [1]. The last two decades has seen the rise of scientific studies of virtual worlds and their potential connection to the real-world [2–6]. Data extracted from virtual worlds allow researchers to test theories on human behaviour in highly controlled environments, for example, sociological behaviour [5, 7], human mobility [8], international relations [9, 10] and economic models [6, 11–13]. A major question, however, is whether the

https://www.ccpgames.com/). We have a contract with Crowd Control Productions that grants us access to the data for non-commercial scientific research. For further requests please contact KS (Koen.Schoors@UGent.be). Authors may also request these data directly from Crowd Control Productions (info@ccpgames.com). The data for the real-world social and economic activities (such as the Global Terrorist Index, the Global Peace Index, the Consumer Price Index, the Real Effective Exchange Rate, the Unemployment Rate) used in the presented analysis are publicly available and references are given in the text.

**Funding:** This research was supported by the Research Foundation Flanders (FWO-Flanders) under Grant Number G018115N and G015617N, and by the Bijzonder Onderzoeksfonds from Ghent University under Grant Number BOF2452014000402. The funders had no role in study design, data collection and analysis, decision to publish, or preparation of the manuscript.

**Competing interests:** The authors have declared that no competing interests exist.

data extracted from virtual worlds are representative enough to study real-world socioeconomic phenomena and whether they capture real-world human behaviour. In a related content many other works have addressed the issue whether digital data can be exploited to assess specific socioeconomic phenomena. Examples include the use of online posts and mobile communications to infer individual economic status [14], and inferring socioeconomic status from large-scale online data sources [15].

We now provide some examples of how social and economic actions in virtual worlds can be used to study socioeconomic phenomena. World of Warcraft (WoW) provides a popular virtual (online) world in a fantasy setting where players can either fight each other or collaborate to meet the game's preset challenges. Martoncik [16] observed that WoW players were on average more social in the game than in the real-world. Through interactions in the game, players develop different feelings and generally experience less loneliness and social anxiety compared to those feelings in the real-world. The difference is more prominent for players who prefer to operate in in-game social groups. Zhang and Kaufman [17] studied the social interactions in WoW and the socio-emotional well-being of players. They found that adult players who are part of a club inside WoW can develop meaningful online relationships including friendship with other players. Destiny is also a virtual (online) world in a futuristic setting, where players can play either individually or in groups (with real-world friends or strangers) to complete a number of challenges designed by the game developers. Destiny was used by Perry et al. [18] to analyse the impact of (in- and out-game) relationships between friends and strangers in the in-game behaviour of players. Pardus is another virtual (online) game that can be accessed via an Internet browser. It represents a futuristic universe where players compete for the limited space whilst interacting in various ways (including fighting) with other players. It has been extensively used for scientific research [4, 5, 8, 19, 20]. In particular, Szell et al. [19] analysed the population of players and the different interaction channels between them and found that in-game communication, friendship, and animosity interaction networks had non-trivial structural properties. Using both the friendship and enmity networks, they found strong evidence for a generative mechanism known as social balance theory [21, 22] that is also found in real-world networks [23–25]. The typical number of social interactions per player in Pardus [26] is similar to Dunbar's number, in the range 100-250, that is a suggested cognitive limit to the number of people with whom one can maintain stable social relationships in the real-world [27, 28]. Thurner et al. in [20] studied sequences of in-game actions in Pardus and observed a higher probability of "positive" ("negative") action if the player had received a "positive" ("negative") action in the preceding time step, suggesting the existence of reciprocal in-game behaviour.

EVE Online is a sandbox massive multi-player on-line game (MMOG) [29] developed by CCP games [29] where over half a million players fight, trade, collaborate, and explore a futuristic galaxy. The players' actions are recorded with a high level of precision and detail in space and time. Feng et al. [30] have studied the temporal evolution of in-game activity and population for three consecutive years. They found daily cycles and weekly periodicity in the number of social connections of players. EVE Online provides a player-driven complex economy as all items need to be produced from raw materials and traded by players with minimal intervention of the game developers. Hoefman et al. [13] have studied the price of the items in the EVE Online free market and found empirical evidence in support of the theory of hedonic pricing. Hedonic pricing theory proposes that the price of an item in a homogeneous class of goods can be expressed as a function of the prices of the different functional characteristics of that item [31]. Although EVE players have full information about functional characteristics and strong incentives for rational behaviour, they are not only willing to pay for functional characteristics but also for social characteristics of in-game items [13]. Using EVE data, Carter [32]

has conducted a study of the use of propaganda during war times as an essential factor for maintaining the morale. Belaza et al. [9, 10] have developed a framework based on statistical physics and structural balance to model international relationships. The model was validated against the network of relationships between alliance groups in EVE Online and international relationships between countries during the Cold War era that have similar network structures. The authors provided evidence that a virtual world can be used as a cleaner and a more extensive laboratory than the data from the real-world for evaluating generative mechanisms for networks of relationships observed in the real-world.

In this paper we study the virtual-world Eve Online [29]. We use data on daily social and economic activities of players to study whether the real-world socioeconomic environment in which players live permeates their in-game behaviour. The focus of our work is on connections between real-world and in-game socioeconomic behaviour at the country level. To this end, we time-aggregate over the data of all individual players from a particular country. We first propose a methodology to generate from a time series of selected player in-game activities, country profiles for average in-game player activity. Next we study the statistical relation between in-game and real-world behaviour using international indexes at the country level. Our research contributes to understanding to what extent virtual worlds can be seen as effective and reliable proxies that accurately reflect the country variation in real-world environments.

## Materials and methods

In this section, we first introduce EVE Online game, the relevant data of players in-game activities, and the real-world socioeconomic indexes that will be used. Then, we introduce a methodology to aggregate player behaviour and trade activity at the country level.

### EVE Online

EVE Online is a sandbox massive multi-player on-line game (MMOG) [29] developed by CCP games [29]. To play EVE Online, a player needs to pay a subscription fee of 15 USD or 15 EUR per month, or spend the equivalent amount by buying an item (called PLEX) inside the game. Once the item is activated and consumed, the subscription extends for another month. The mere fact that there is a subscription fee in combination with the technological requirements impacts the distribution of players over the different countries. Per capita we observe that there tends to be more players from high-income countries than from low-income countries. For countries with more than 100 accounts, the number of accounts corresponds with a fraction of the total population that varies from the subpercent level to about 8%. There is one exception to this: for Iceland, where the headquarter of CCP games is located, the number of accounts corresponds with about 30% of the total population. There are over half a million players who fight, trade, collaborate, and explore a futuristic galaxy. Players can choose to engage in different activities, ranging from mining asteroids to waging war. For example, the distribution of raw materials is pre-defined by the game developers but the extraction, processing and manufacturing of items, and transport and trading of goods are made and controlled by players. The player's experience is shaped by the activities of the other players. Altogether, this dynamics gives rise to an emergent economy with specialisation of players and intricate economic interactions. Inter-player interactions also play a role at other levels since they create complex social hierarchies used to control territories and exploit their resources. Not least, these structures and frequent inter-player interactions generate a complex world with espionage and game-wide wars as emergent social phenomena. The analysis presented in this work

is based on collected EVE Online data from December 2011 to December 2016 (61 months) for the trade activity and the year 2016 for the social and economic activity.

## Measures of in-game social and economic activity

We select EVE activities and extract data that are most relevant to study social behaviour, aggressiveness, and production of individual players (Table 1). To capture the in-game social behaviour of a player, we define two variables: (i) "*Added As Negative*" and (ii) "*Added As Positive*" encoding whether a particular player gets tagged as friend or enemy, respectively, during inter-player contacts. The difference between these two variables is a measure of in-game pro-social behaviour ("*Reputation*"). The number of times a player gets involved in virtual combats with other players, both as aggressor or defender, gives the in-game aggressiveness of a player. *Aggression* is thus the number of "*Engaged Attack*" minus "*Engaged Defense*". In-game aggressiveness is also captured by the number of events involving killing of a Non-Player Character (NPC), which is a ship controlled by the game ("*Kill NPC*"). To measure in-game production (and recycling) activities of the individual players, we use three variables: the actual production of items ("*Production*"), the extraction of raw materials from Asteroids ("*Mining*") and extraction of material from destroyed ships ("*Salvaging*").

## Measures of real-world social and economic activity

To measure aggressiveness in the real-world at the country level, we use the Global Terrorist Index (GTI) and the Global Peace Index (GPI) from the year 2016. The GTI [33] is proposed as a measure of the impact of terrorism in 196 different countries and comes from the Global Terrorist Database curated by the Institute for Economics & Peace. The GPI [34], published by the same institute, combines data on terrorist attacks and other indicators, including for example, criminality rates and military expenses. A low value of GPI indicates a higher level of peace.

To measure real-world socio-economic characteristics of each country, we use the Consumer Price Index (CPI), the Real Effective Exchange Rate (REER), and the Unemployment

**Table 1. In-game activities representing the levels of social interaction, of aggressiveness, and of production activity of players.**

| Variable | Category | Description |
|---|---|---|
| *Added As Negative* | social interaction | Interacting players can mark the counter-party as friend, enemy or neutral. |
| *Added As Positive* | social interaction | This mark is visible whenever players meet. |
| *Reputation* | social interaction | (*Added As Positive*)—(*Added As Negative*) |
| *Engaged Attack* | aggressiveness | Attack another player. |
| *Engaged Defense* | aggressiveness | Being attacked by another player. |
| *Aggression* | aggressiveness | Conflict against others: (*Engaged Attack*)—(*Engaged Defense*). |
| *Kill NPC* | aggressiveness | Destroy ships controlled by the game, usually so-called pirates. |
| *Production* | production activity | Produce an item, using other items as input. |
| *Mining* | production activity | Extract materials from asteroids. |
| *Industrial* | production activity | *Production + Mining* |
| *Salvaging* | production activity | Recover materials from destroyed ships. |

rate (UNEMP) from the World Bank for the year 2016 [35]. CPI is an inflation index measuring the evolution of a weighted average price level of a fixed basket of basic consumer goods and services like food, ICT, medical care and housing. REER is an exchange rate index describing the strength of the local currency with respect to the rest of the world. It is a weighted average of all exchange rates of the local country currency to the currencies of all the country's trading partners, weighted by the trade volume. UNEMP is the fraction of the working age population that is unemployed and looking for a job.

## In-game country profile

We construct an activity-based in-game profile per country from the selection of eight independent key activities listed in Table 1. A country's profile is based on the time and player average profile of the players that come from that particular country. Upon determining the time series of the players' in-game profiles we face the challenge of heterogeneity in the time scales of the selected in-game activities. For example, marking someone as a friend is an instantaneous event whereas the process of mining an asteroid can take up to half a minute. Data for the individual players are aggregated on a weekly basis, creating a specific profile per week for all players and weeks in 2016. To deal with different scales and metrics, we define an adimensional "activity coefficient" that combines in-game events and is representative for the activity level of each player during a given week. The ratio of the action to the normal activity determines the adimensional coefficient. The temporal evolution of these profiles could for example be used to study in how far players adapt their in-game behaviour in response to an out-game stimulus. These studies are beyond the scope of the current work as our focus is on collective socioeconomic behaviour at the country level.

For the country profile, we take the average of all the profiles of players belonging to that country and normalise by the standard score of all countries with at least 15 players in EVE online. The threshold of a minimum of 15 players is a trade-off between data cleanliness and country coverage. We wish to have as many countries as possible in our analysis. We have checked that noisy country profiles are generated when including countries with only a few players. The required minimum number of players (= 15) in each country is set to a level to keep a significant sample of countries (106 countries) for analysis. We have checked that increasing the threshold from 15 to a higher number does not markedly modify the results whilst reducing the country coverage in our analysis (See In-game player behaviour and the real-world socioeconomic environment).

For the sake of clarity we provide a step wise description of the data curation process that allowed us to create country profiles from the recorded players' in-game activities during the year 2016.

Step 1: For each player and week, we sum the number of times they perform each action. The variables *Production*, *Mining* and *Salvaging* involve the production of an item and are measured through monetary value instead of counting the number of times that the activity has been completed. The produced value is the sum over the items whereby the cost of each item is determined by that week's average market price. For example, a player mining two items with a market price of 5000 ISK and three items with a market price of 1000 ISK has a *Mining* value of 13000 ISK for that week. At the end of this step we possess a weekly player profile for all studied actions.

Step 2: We remove the activities from "sporadic players", i.e. players that have no recorded activities for all weeks of the covered time period.

Step 3: To deal with different scales and metrics, for each player-measure combination we define an adimensional "activity coefficient" for each week. To this end, for each measure and player we determine the ratio of the player's measure of that week relative to the cross-player and cross-week average. In computing those averages all zeroes are removed. The resulting profile of each player provides for each measure information on how more or less frequent the player performs the action in comparison with the "average player". For example, if all players killing NPCs kill on average 200 times per week, and a specific player kills 500 times in a particular week, the player's activity is recorded as "*Kill NPC* 500/200 = 2.5" for that particular week. For each week during the period analysed we determine the profile for all individual players and measures.

Step 4: To create a country profile for a particular measure, we average the weekly profile discussed in Step 3 over all its players.

Step 5: The country-based measures for the in-game activities are re-scaled to zero mean and unit variance (Z-score).

## Measures of in-game trade activity

The EVE market operates according to a double auction system. Players can set a sell or buy order of a specific item for a certain price. They can also see the full list of active orders and opt to fulfill them. For each country $c$ and item $i$, we determine the monthly sell price $P_{Si}^c$, the monthly buy price $P_{Bi}^c$, and the monthly volume of sell and buy orders, respectively, $V_{Si}^c$ and $V_{Bi}^c$. We collect the data for $M = 106$ countries $c$ with more than 15 players and for 6,882 unique items $i$. Table 2 shows the in-game measures of trade activity for each player for a specific country. All measures include a sum over all the different items. The mean for each item $i$ is taken over $M$ countries, for example, for buy price $\langle Bi \rangle_c = \sum_{c=1}^{c=M} \frac{P_{Bi}^c}{M}$, with equivalent definitions for the other variables.

## Results

In this section, we first present the activity-based in-game country profiles and then perform a regression analysis of the socioeconomic behaviour of players inside the game in comparison to the their country-of-origin in the real-world.

**Table 2. Measures of in-game trade activity of individual players grouped according to country-of-origin.**

| Variable | Definition | Description |
|---|---|---|
| *Buy Price* | $\sum_i \dfrac{P_{Bi}^c - \langle P_{Bi} \rangle_c}{\langle P_{Bi} \rangle_c}$ | Buy price in country $c$ relative to the average over all countries. |
| *Bid Ask* | $\sum_i \dfrac{P_{Si}^c - P_{Bi}^c}{P_{Si}^c + P_{Bi}^c}$ | Bid-ask spread in prices for country $c$. |
| *Buy Volume* | $\sum_i \dfrac{V_{Bi}^c - \langle V_{Bi} \rangle_c}{\langle V_{Bi} \rangle_c}$ | Volume per transaction of buying orders in country $c$ relative to the average across countries. |
| *Bid Ask Volume* | $\sum_i \dfrac{V_{Si}^c - V_{Bi}^c}{V_{Si}^c + V_{Bi}^c}$ | Bid-ask spread for the traded volume per transaction in country $c$. |
| *Bid Ask Transactions* | $\sum_i \dfrac{N_{Si}^c - N_{Bi}^c}{N_{Si}^c + N_{Bi}^c}$ | Bid-ask spread in the number of transactions in country $c$. |

$P_{Bi}^c, V_{Bi}^c, N_{Bi}^c$ stand respectively for price, volume, and number of transactions of buy orders for item $i$ in country $c$. $P_{Si}^c, V_{Si}^c, N_{Si}^c$ are the equivalent for sell orders.

**Table 3. Country profiles for selected countries.**

| Country | Russia | Belarus | Ukraine | Canada | France | UK | Germany | Austria | Japan | Philippines |
|---|---|---|---|---|---|---|---|---|---|---|
| *Added As Negative* | 0.177 | 0.357 | 0.325 | -0.024 | 0.133 | 0.209 | 0.204 | 0.198 | 0.008 | 0.323 |
| *Added As Positive* | 0.291 | 0.314 | 0.315 | 0.068 | 0.283 | 0.232 | 0.110 | 0.136 | -0.008 | 0.339 |
| *Engaged Attack* | 0.384 | 0.444 | 0.439 | 0.355 | 0.389 | 0.413 | 0.360 | 0.384 | -0.297 | 0.012 |
| *Engaged Defense* | 0.560 | 0.684 | 0.670 | 0.512 | 0.542 | 0.570 | 0.457 | 0.476 | -0.423 | 0.324 |
| *Kill NPC* | 0.849 | 1.207 | 1.556 | -0.230 | -0.095 | -0.169 | 0.055 | -0.068 | -0.079 | 0.030 |
| *Production* | 0.559 | 1.344 | 1.121 | 0.015 | 0.010 | 0.054 | 0.275 | -0.033 | 0.405 | -0.293 |
| *Mining* | 0.610 | 0.856 | 0.702 | 0.335 | 0.408 | 0.322 | 0.617 | 0.590 | 0.356 | 0.200 |
| *Salvaging* | 0.615 | 0.691 | 0.597 | 0.255 | 0.339 | 0.275 | 0.389 | 0.310 | 0.171 | 0.515 |
| Cosine similarity* | 0.950 | | -0.982 | | 0.792 | | 0.823 | | -0.132 | |

*Cosine similarity varies from −1 (low similarity) to + 1 (high similarity).

## In-game country profiles and trade activity

The in-game country profile is generated by combining a selected number of in-game activities capturing the social interactions, aggressiveness, and production of players. Table 3 shows the country profiles of selected countries and the cosine similarity between pairs of countries for eight in-game activities (See section Materials and methods for details). The cosine similarity is calculated using the hyper-dimensional vectors that correspond to the country profiles based on all eight in-game activities.

We analysed the similarity between country profiles using the subset of countries with more than 100 players (62 countries). Fig 1 shows the cosine similarity of pairs of countries. The countries are first clustered via an iterative algorithm that at each time step aggregates the two closest clusters (See function *hierarchy.linkage* in the Python scipy.cluster module and [36, 37] for more details of the algorithm) and then ordered in clusters with similar profiles. Fig 1 shows one cluster of countries (upper-left part of the matrix) with highly similar profiles, with Oceanic and Asian countries a bit over-represented, and another cluster of Eastern Europe countries, i.e. Ukraine, Russia, Belarus, Moldova, Bulgaria, Lithuania, among others. Furthermore, France and the United Kingdom have similar profiles, as well as Germany and Austria, suggesting that geography may be relevant. Canada and Ukraine are the countries with highest profile difference (cosine similarity is −0.98). The calculated similarities indicate the robustness of our method.

Using the transaction data from December 2011 to December 2016, we create monthly (61 months) measures of in-game trade activity of individual players grouped according to country-of-origin for the 106 countries with more than 15 players (Table 2). The data is normalised using the average and the standard deviation (Z-score). We also winsorize the data by removing 1.5% of the outliers, i.e. all data points larger than 3 standard deviations. Table 4 shows the summary statistics for each trade activity.

## In-game player behaviour and the real-world socioeconomic environment

In this section we study the relation between real-world aggressiveness at country-level and in-game social interactions, aggressiveness, and economic activities of players at the level of their country of origin. Previous studies have suggested that the real-world aggressiveness in a country as defined by the Global Terrorist Index (GTI) correlates with a variety of social, economic, and geographical measures, like the linguistic fractionalisation, the GDP, and the Human Development Index (HDI) [38, 39]. As players perform social and economic activities in EVE

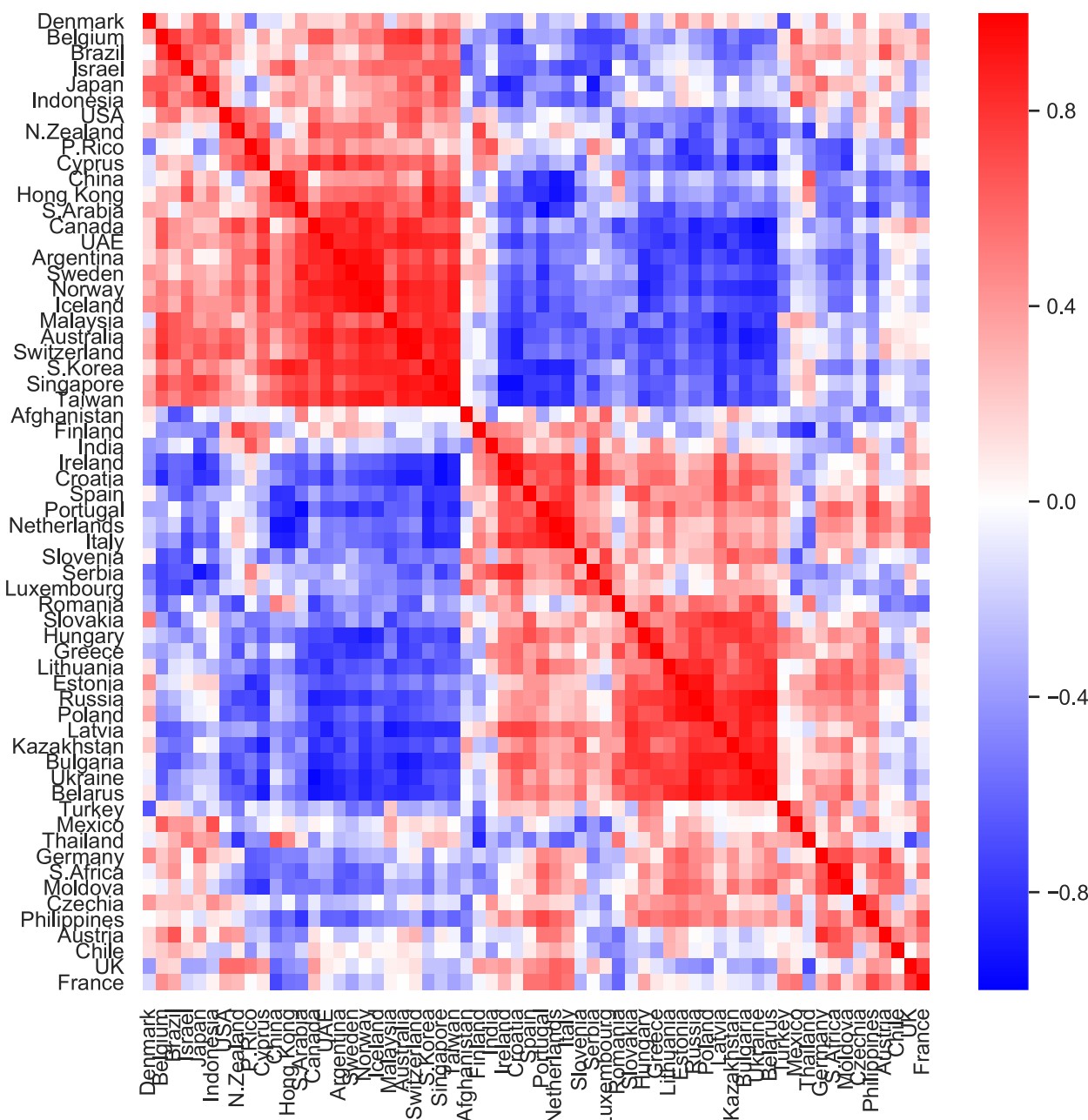

**Fig 1. In-game similarity between countries.** Cosine similarity between in-game socioeconomic profiles of countries with more than 100 players (+ 1 indicates high similarity and −1 low similarity). The profiles are represented as hyper-dimensional vectors and are obtained with the eight independent variables in Table 1, after averaging over all players for each country (See Materials and methods).

Online, we hypothesise that the players' in-game activities and attitudes reflect their real-world socioeconomic environment.

For both the GTI and the GPI, and for all the in-game socioeconomic activities in Table 1 (See Materials and methods), we propose the following three models:

- Model I: The minimal model that correlates real-world aggressiveness with aggressive in-game behaviour. These models are GTI I and GPI I.

**Table 4. Statistics of in-game trade activities at the country level.**

|  | count | mean | std | min | 25% | 50% | 75% | max |
|---|---|---|---|---|---|---|---|---|
| *Buy Price* | 2113 | 0.159 | 0.331 | -0.823 | -0.005 | 0.099 | 0.235 | 2.946 |
| *Bid Ask* | 1614 | -0.088 | 0.183 | -0.991 | -0.158 | -0.079 | -0.006 | 0.895 |
| *Buy Volume* | 2113 | -0.052 | 0.426 | -1.000 | -0.283 | -0.126 | 0.065 | 2.981 |
| *Bid Ask Volume* | 1614 | -0.067 | 0.229 | -1.000 | -0.155 | -0.041 | 0.029 | 1.000 |
| *Bid Ask Transactions* | 1614 | -0.025 | 0.209 | -0.967 | -0.132 | -0.027 | 0.056 | 0.976 |

- Model II: An extension of Model I variants that also includes pro-social in-game behaviour (i.e. *Reputation*). These models are GTI II and GPI II.

- Model III: An extension of Model II variants that adds the measures of productive in-game behaviour (*Industrial*, *Salvaging*). These models are GTI III and GPI III.

Table 5 shows the results of Ordinary Least Squares (OLS) regressions for all models. *Aggression* measures the likelihood that players of a given country are the aggressors in the fights they have with other players. Positive *Aggression* indicates that players of this country tend to be the first shooter in the game fights they get involved. *Industrial* is a country-level measure of *Production* and *Mining* activities. *Salvaging* is a country-level measure of extraction of materials from destroyed ships and is thought of as productive (recycling) behaviour in the game. For countries with more than 15 players, there is correlation between the real-world aggressiveness (GTI and GPI) and in-game player behaviour. The linear regression for the

**Table 5. Real-world aggressiveness regression results per country for year 2016.**

|  | GTI I | GTI II | GTI III | GPI I | GPI II | GPI III |
|---|---|---|---|---|---|---|
| *Aggression* | -0.71** | -0.70** | -0.98*** | -0.15*** | -0.15*** | -0.22*** |
|  | (0.28) | (0.28) | (0.29) | (0.05) | (0.05) | (0.06) |
| *Kill NPC* | 0.42 | 0.42 | 1.08** | 0.07 | 0.08 | 0.19*** |
|  | (0.33) | (0.34) | (0.41) | (0.06) | (0.06) | (0.07) |
| *Reputation* |  | -0.20 | 0.39 |  | 0.11** | 0.21*** |
|  |  | (0.30) | (0.36) |  | (0.05) | (0.06) |
| *Industrial* |  |  | -0.31 |  |  | -0.04 |
|  |  |  | (0.56) |  |  | (0.10) |
| *Salvaging* |  |  | -0.96** |  |  | -0.19** |
|  |  |  | (0.43) |  |  | (0.08) |
| Intercept | 2.94*** | 2.95*** | 3.32*** | 1.97*** | 1.97*** | 2.02*** |
|  | (0.29) | (0.29) | (0.64) | (0.05) | (0.05) | (0.11) |
| No. observations | 71 | 71 | 71 | 88 | 88 | 88 |
| $R^2$ | 0.11 | 0.12 | 0.21 | 0.11 | 0.15 | 0.23 |
| Adjusted $R^2$ | 0.09 | 0.08 | 0.14 | 0.09 | 0.12 | 0.18 |
| Min Eigenval | 5.33e+01 | 5.31e+01 | 7.72e+00 | 7.01e+01 | 6.98e+01 | 1.00e+01 |
| Condition number | 1.22e+00 | 1.23e+00 | 5.04e+00 | 1.12e+00 | 1.13e+00 | 4.84e+00 |
| AIC | 3.32e+02 | 3.34e+02 | 3.30e+02 | 1.27e+02 | 1.25e+02 | 1.21e+02 |

Ordinary Least Squares (OLS) regressions. Standard errors in parentheses.

* $p < .1$,

** $p < .05$,

*** $p < .01$.

Global Terrorist Index (GTI) has a lower $R^2$ than for the Global Peace Index (GPI) (Table 5), that takes into account more variables than GTI. GTI only includes intentional acts of violence or threats of violence by a non-state actor (See [33]). On the other hand, GPI includes more indicators as for example the jail population per capita, the size of the armed forces per capita, the number of homicides per capita, as well as the Terrorist index (See [34] for the methodology to construct the indexes).

Based on the AIC (Akaike information criterion), none of the models can be discarded. For all tested models, the coefficient of the in-game aggressiveness against non-player characters (*Kill NPC*) is positive, though only significantly in models GTI III and GPI III. In-game aggressiveness against NPCs is thus related to higher real-world aggressiveness. The opposite is also true for aggressiveness against fellow players, i.e. *Aggression* is negatively correlated with both the GTI and the GPI indexes in all our models. These results suggest that in-game aggressive behaviour against other players is not a good measure of real-world aggressiveness in the country of origin of these players. Although players from unsafer countries display on average more in-game aggressiveness against non-player characters, they also show on average less in-game aggressiveness towards other players than players from safer countries. One interpretation is that in-game fighting with actual players is in fact a social activity, while systematically killing of non-players characters reflects real world aggressiveness. Another interpretation is that in-game aggressiveness against other players may act as a safety valve for inherent aggression, making the real world a safer place. We turn back to this interpretation in the discussion.

Pro-social behaviour is given by the *Reputation* that is measured as *Added As Positive* minus *Added As Negative*. Positive country's *Reputation* indicates that other players mark players from this country more often as friends than enemies. Pro-social behaviour is positively correlated with the GPI, implying that players from more aggressive and less peaceful countries tend to have more friends than enemies in the virtual world, suggesting again that the virtual world may to some extent act as a safety valve: if you life in a country where making real friends is hard, you can still make then on line.

We have performed robustness checks of the above results. First, we have divided our data into two groups containing respectively 80% and 20% of the data. The linear regression model estimated with the 80% group shows results qualitatively similar to those of the 20%-group. The results from the full data set and the 80%-group are also consistent. In addition, we have tested the imposed minimum number of players per country required for inclusion in the sample (as discussed above we set the threshold at 15 players per country). Imposing a stricter minimum of 25 players per country (and thus adding less countries in the sample) produces results completely in line with those of Table 5 with comparable values for $R^2$. On the other hand, relaxing the inclusion constraint to a minimum of 10 players (consequently adding countries with very few players to the sample) introduces very noisy observations in the sample and produces results that are more difficult to interpret. For example, comparing the results for the threshold value of 10 to those for a threshold value of 15, the value of $R^2$ for the GPI III model is reduced by about a factor of two and also the observed signal for *Reputation* weakens substantially.

## In-game trading activity and real-world economic outcomes

In this section we study the relation between real-world economic outcomes at the country level and measures of in-game trade activity (market behaviour) at the level of the country of origin of the player. Since a player can pay the subscription either with real-world money or buy in-game items from other players, those earning enough in-game currency can pay the full subscription without spending any real-world money. Through the subscription cost in the

**Table 6. Real-world economic outcomes regression results per country for year 2016.**

|  | CPI | REER | UNEMP |
|---|---|---|---|
| *Buy Price* | 0.37 | -2.51** | -1.39*** |
|  | (3.74) | (1.03) | (0.47) |
| *Bid Ask* | 2.48 | 3.70** | -2.64*** |
|  | (5.37) | (1.51) | (0.68) |
| *Buy Volume* | -1.18 | -1.78*** | -0.68** |
|  | (2.46) | (0.66) | (0.32) |
| *Bid Ask Volume* | -15.77*** | 0.39 | -0.37 |
|  | (4.18) | (1.16) | (0.53) |
| *Bid Ask Transactions* | 3.20 | 1.93 | 1.17** |
|  | (4.56) | (1.23) | (0.58) |
| Intercept | 108.90*** | 102.27*** | 8.17*** |
|  | (2.09) | (0.56) | (0.26) |
| No. observations | 1500 | 1499 | 1554 |
| $R^2$ | 0.02 | 0.04 | 0.02 |
| Adjusted $R^2$ | 0.02 | 0.03 | 0.02 |
| Min Eigenval | 4.34e+01 | 3.87e+01 | 4.38e+01 |
| Cond. number | 2.29e+02 | 2.43e+02 | 2.31e+02 |

Ordinary Least Squares (OLS) regressions. Standard errors in parentheses.

* $p < .1$,

** $p < .05$,

*** $p < .01$.

local real-world currency, price variations and unemployed risk in the real-world can all be transmitted to the in-game financial behaviour. We hypothesise that unemployment risk will encourage players to be more cautious in their in-game transactions, as the opportunity cost of real-world money has increased whereas the cost of spending in-game time has decreased. We also hypothesise that exchange rates affect the ratio between the value of in-game currency and real-world currency. Players may react to a real-world depreciation by being more eager to pay for their subscription with money earned in the game, for example by trading more actively or by demanding higher prices for selling in-game goods. Our third hypothesis is based on the fact that inflation creates a connection between real-world money and real-world goods. Therefore, players who experience inflation in the real-world may be inclined to mimic this behaviour in the virtual world by increasing in-game prices (both buy and sell prices). We propose a single regression model for each variable CPI, REER, and UNEMP, taking into account all the in-game trade activities in Table 2 (See Materials and methods).

Table 6 shows that all three regression analysis have low explanatory power (low $R^2$). Players from countries with higher consumer prices inflation in real-world items (CPI) tend to buy more than sell in-game (higher *Bid Ask Volume*). This result suggests that real-world inflation experiences in the country-of-origin are transmitted into in-game inflation expectations. The real effective exchange rate is negatively correlated with the in-game *Buy Price* and *Buy Volume*, but positively correlated with the bid-ask spread (*Bid Ask*). That means if a country's real currency depreciates, in-game money becomes relatively more expensive for the players of this country, urging them to buy smaller quantities at lower prices and to trade more efficiently. Finally, the level of unemployment is strongly related to most of the studied in-game trading behaviours. Players from countries with higher levels of unemployment not only buy less (*Buy*

*Volume*), at cheaper prices (*Buy Price*) and more efficiently (*Bid Ask*), but also have more sell transactions relatively to buy transactions (*Bid Ask*). These findings suggest that players of countries with high unemployment attach a relatively higher opportunity cost to real-world money than to real-world time, increasing their willingness to work in the game to earn in-game money necessary to pay for their subscription, since real-world money is limited or nonexistent.

To check the robustness of our results, here we also randomly divided our data in two sets with 80% and 20% of the original data. The estimations using the 80% sample yields results consistent with those given by the full sample. Estimates using the remaining 20% of the data are also inline to the full data. The root-mean-square error (RMSE) of the estimations on the two samples differ less than 10%.

## Discussion

EVE Online is a paid multi-player online game where over half a million players fight, trade, collaborate, and explore a futuristic galaxy. It is not only a social but also a economic game where one of the main activities is trade. In this paper, we propose a series of socioeconomic hypotheses and test them using EVE Online data to find whether in-game player behaviour is connected to real-world human behaviour and to the socioeconomic context in which players live.

We first propose a methodology to combine data from multiple players and to create a representative country profile containing the average player behaviour. This is a fundamental step because individual player data are noisy and difficult to link to individual real-world behaviour in this context. The country-level analysis also helps to understand particular socioeconomic aspects of societies.

Our analysis indicates a negative relation between in-game aggressiveness against other players and real-world aggressiveness in the players country of residence, as measured by the Global Peace Index and Global Terrorism Index. This result suggests that players experiencing violence in daily life have a tendency to be less aggressive to fellow players in comparison to players who are less exposed to violence and its consequences to society. This unexpected result is in line with the findings of the study of [40] where it was observed that in the USA violent crime was more likely to show decreases instead of increases in response to the release of violent video games. One possible explanation for this reduction in violence put forward in [40] is that playing violent video games works as a safety valve for inherent aggression and thus leads to a catharsis. In other words, when people play violent video games, they are able to release their aggression in the virtual world instead of in the real world. We further find that players from less peaceful countries tend to make and maintain more friends than enemies in the game relatively to those living in less violent countries. This result indicates that trust and confidence in the virtual world may act as a substitute for positive real-world social experiences. We have observed however a positive relation between in-game aggressiveness against non-player characters and real-world aggressiveness. In this case, players from more violent countries tend to act more violently against non-human characters than those living in peaceful countries.

The analysis of trading patterns indicate a significant, though relatively small, correlation between in-game trading and real-world variables reflecting the macroeconomic context where players live. We find that higher unemployment rates and weaker currencies in the country of residence are associated to more efficient and money-conscious in-game trading behaviour and market dynamics. Not least, the less attractive economic environment of certain countries encourages players from these countries to earn more in-game money. This can be

partially explained by the fact that players can use in-game money to pay for their continuous membership on EVE Online.

In this paper, we have found evidence of correlations between social and economic indicators in the real world and the players behaviour in EVE Online, a game simulating a fantasy galaxy with real-world inspired trading and social dynamics. While correlations do not necessarily imply causality, it is intuitive to expect in our study that due to the relative dimensions (i.e. population sizes of both worlds), the country-of-origin of players affect their in-game behaviour and not vice-versa, as in online vs. offline feedback dynamics in specific contexts [41]. Our statistical models could be further improved to better explain the variation of the real-world socioeconomic indicators, their temporal evolution, and the mechanisms driving the influence and inter-relations between players in-game. The study of the temporal evolution of some quantities and reactions to real-world events could be done for countries with a large amount of players. Future research directions also include similar analysis at the individual player level using online questionnaires. Overall, our study provides evidence on the in-game impact of the real-world socioeconomic environment and socio-cultural norms in which players are immersed and experience daily-life. These results also support previous research encouraging the use of virtual worlds as realistic in-silico environments to perform controlled experiments and test relevant social and economic theories.

## Supporting information

**S1 File. Data used to compute the country profiles contained in Fig 1.**
(CSV)

## Acknowledgments

We would like to thank Milan van den Heuvel for his invaluable input.

## Author Contributions

**Conceptualization:** Andres M. Belaza, Jan Ryckebusch, Koen Schoors.

**Data curation:** Andres M. Belaza, Benjamin Vandermarliere.

**Formal analysis:** Andres M. Belaza.

**Funding acquisition:** Jan Ryckebusch, Koen Schoors.

**Methodology:** Jan Ryckebusch, Koen Schoors, Benjamin Vandermarliere.

**Project administration:** Jan Ryckebusch, Koen Schoors.

**Software:** Andres M. Belaza, Benjamin Vandermarliere.

**Supervision:** Jan Ryckebusch, Koen Schoors, Luis E. C. Rocha, Benjamin Vandermarliere.

**Validation:** Koen Schoors.

**Visualization:** Andres M. Belaza.

**Writing – original draft:** Andres M. Belaza, Jan Ryckebusch, Koen Schoors, Luis E. C. Rocha.

**Writing – review & editing:** Jan Ryckebusch, Koen Schoors.

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
