## [Decision Letter · Decision Letter 0]

30 Jul 2020

PONE-D-20-14789

On the connection between real-world circumstances and  online player behaviour: the case of EVE Online

PLOS ONE

Dear Dr. Ryckebusch,

Thank you for submitting your manuscript to PLOS ONE. After careful consideration, we feel that it has merit but does not fully meet PLOS ONE’s publication criteria as it currently stands. Therefore, we invite you to submit a revised version of the manuscript that addresses the points raised during the review process.

Please note that references suggested by reviewers could be ignored.

We look forward to receiving your revised manuscript.

Kind regards,

Jichang Zhao, Ph.D.

Academic Editor

PLOS ONE

Journal Requirements:

2. Please include a brief description in your methods section of how you accessed/collected the data, and how your full dataset can be accessed by future researchers.

3. Please remove your figures from within your manuscript file, leaving only the individual TIFF/EPS image files, uploaded separately.  These will be automatically included in the reviewers’ PDF.

'The funders had no role in study design, data collection and analysis, decision to publish, or preparation of the manuscript.'

Additional Editor Comments (if provided):

Reviewers' comments:

Reviewer's Responses to Questions

**Comments to the Author**

1. Is the manuscript technically sound, and do the data support the conclusions?

Reviewer #1: Yes

Reviewer #2: Yes

2. Has the statistical analysis been performed appropriately and rigorously? 

Reviewer #1: Yes

Reviewer #2: Yes

3. Have the authors made all data underlying the findings in their manuscript fully available?

Reviewer #1: Yes

Reviewer #2: Yes

4. Is the manuscript presented in an intelligible fashion and written in standard English?

Reviewer #1: Yes

Reviewer #2: Yes

5. Review Comments to the Author

Reviewer #1: Review report for “On the connection between real-world circumstances and online player behaviour: the case of EVE Online” by Belaza et al.

This paper studies the connection between real-world circumstances and behaviors of online players by a case study of the EVE Online game. Specifically, the authors explored the influence of the real-world context in which players live on their in-game behavior. They found a negative relation between in-game aggressiveness and real-world aggressiveness in the players country of residence, showing that players experiencing violence in daily life have a tendency to be less aggressive in online games. Moreover, players from more peaceful countries tend to maintain more friends relatively to those living in more violent countries. The authors also found some correlations between in-game trading and real-world variables reflecting the macroeconomic context where players live, for example, there are correlations between unemployment rates and money-conscious in-game trading behavior.

Understanding the connections between real-world circumstances and online behaviors is an interesting and important issue, which has implications for better understanding human behaviors and designing better experiments with a variety of settings. I think this paper considers an interesting research question, the results and analyses are convincing, and the structure of the paper is good. In my view, this paper deserves a final publication. In the following, I would like to provide some suggestions and comments, which may be helpful for the authors to further strength their paper.

1. In the abstraction section, the authors presented that “Virtual worlds involving social and economic interactions are particularly useful to test social and economic theories”. I think this is particularly interesting considering that analyzing behaviors in virtual worlds has relatively low cost and virtual worlds are more flexible in implementing experiments to test social and economic theories if they are well representations of offline worlds. The authors also mentioned in the first paragraph of the introduction section that “A major question, however, is whether the data extracted from virtual worlds are representative enough to study real-world socioeconomic phenomena and whether they capture real-world human behaviour”. There are some previous works that can better support the authors’ narratives in related content, for example, using online posts and mobile communications to infer individual economic status [Luo, S., Morone, F., Sarraute, C., Travizano, M., & Makse, H. A. (2017). Inferring personal economic status from social network location. Nature communications, 8(1), 1-7], and infering socioeconomic status from large-scale online data sources [Gao, J., Zhang, Y. C., & Zhou, T. (2019). Computational socioeconomics. Physics Reports, 817, 1-104]. The authors are suggested to cover a relative broad branch of literature and present a more general introduction of the background of this paper.

2. On page 2, the authors presented that “Using EVE data, Carter has conducted a study of the use of propaganda during war times as an essential factor for maintaining the morale [28]”. It would be better to keep the typical citing format as the authors used. That is “Using EVE data, Carter [28] has conducted a study xx”. On page 3, the authors presented that “To play EVE Online, a player needs to pay a subscription fee of 15 USD or 15 EUR per month, or spend the equivalent amount by buying an item (called PLEX) inside the game”. I am wondering how the subscription fee will affect the enrollments of the EVE Online. For some low-income countries, the sampling rate of all population may be different. It would be necessary to present a new figure showing the number of users for each country, and the share to all population in each country. I am wondering if the sampling bias will affect the observations.

3. On page 3, the authors presented that “The analysis presented in this work is based on collected EVE Online data from December 2011 to December 2016 (61 months) for the trade activity and the year 2016 for the social and economic activity”. It seems that the authors used cumulative data across their study, however, I am wondering if there are any temporal changes of these patterns observed in this paper, for example, the correlations between online game behavioral features and real-world economic indicators in the year 2014, and how the results evolve over time. Regarding the two variables: (i) Added As Negative" and (ii) Added As Positive, I think there are also temporal changes, for example, a player can be added as negative and as positive at different periods considering the online game is a dynamitic systems and friendships among players can change quickly. I am wondering how the authors capture these evolving characteristics.

4. On page 4, the authors presented that “The latter three activities result in the generation of an in-game item that we transform into a numeric value using the market prices. Data are aggregated on a weekly basis, creating a specific player profile per week”. I think it remains unclear how the authors did the transformation into a numeric value. The authors are suggested to add more explanations to this point. On page 5, the authors presented that “Finally, the player's profile is normalised by the sum of the eight adimensional measures of the in-game activities”. I am wondering if the method of normalization may affect the results and if the sum of the eight adimensional measures is treating measures with equal weights.

5. In Table 3, the authors presented country profiles for selected countries. I noticed that most of these countries are western countries. Maybe it is better to include some Eastern and Asian countries as well considering a diverse set of cultural backgrounds. In Fig 1, the authors presented the in-game similarity between countries. First, it would be better to show all country names in the x-axis as well. Second, it would be better to reverse the color scheme, where red indicates positive value and blue indicates negative values.

6. In Table 5 and Table 6, it would be better to list Intercept blow other control variables but above the R2. Listing the variables of most interests in the top of the tables is a tradition. Also, No. observations should be listed above R2. In addition, the authors presented some convincing correlations among online behaviors and offline socioeconomic status. Yet, I am wondering if it is possible to identify some causal relationships using instrumental variables. It would be not easy due to the observational data, but I am wondering the possibility. It would be nice to add some related discussions in the discussion section.

7. The authors found that “players from safer countries behave on average more aggressively towards other players in the virtual world than players from unsafer countries”. I am wondering if there are social and psychological explanations for this observation. In the reference section, it would be better to presented the full journal name for PNAS as “Proceedings of the National Academy of Sciences, U.S.A.”. For other references, they are in very good shape of citing format.

Reviewer #2: In this manuscript, authors analyze the influence of the real-world context in which players live on their in-game behavior. This work is of great interests, and the analysis is solid. I would recommend publication after the authors make proper minor modifications.

1 It is better to highlight the contributions of this work and the difference between this work and the previous studies in the introduction.

2 About the in-game country profile, data are aggregated on a weekly basis, creating a specific player profile per week. And the variables on social interaction, aggressiveness and production activity are calculated by the absolute quantity. I think the times players in different countries spend on online game may varies. Therefore, it is better to use the number of activities per unit of online time to calculated the variables, right?

3 “For the country profile, the average of all the profiles of the players belonging to that country is taken and normalized by the standard score of all countries with at least 15 players in EVE online.

”. Please provide more statistic analysis to support the threshold (15 players).

6. PLOS authors have the option to publish the peer review history of their article (what does this mean?). If published, this will include your full peer review and any attached files.

Reviewer #1: No

Reviewer #2: No

---

## [Author Response · Author response to Decision Letter 0]

10 Sep 2020

A rebuttal letter is included in this resubmission.

---

## [Editor Report · Decision Letter 1]

22 Sep 2020

On the connection between real-world circumstances and  online player behaviour: the case of EVE Online

PONE-D-20-14789R1

Dear Dr. Ryckebusch,

We’re pleased to inform you that your manuscript has been judged scientifically suitable for publication and will be formally accepted for publication once it meets all outstanding technical requirements.

Kind regards,

Jichang Zhao, Ph.D.

Academic Editor

PLOS ONE
---

## [Editor Report · Acceptance letter]

29 Sep 2020

PONE-D-20-14789R1 

On the connection between real-world circumstances and  online player behaviour: the case of EVE Online 

Dear Dr. Ryckebusch:

I'm pleased to inform you that your manuscript has been deemed suitable for publication in PLOS ONE. Congratulations! Your manuscript is now with our production department. 

Kind regards, 

on behalf of

Professor Jichang Zhao 

Academic Editor

PLOS ONE